# COVID-19 Associated Pulmonary Aspergillosis in Patients on Extracorporeal Membrane Oxygenation Treatment—A Retrospective Study

**DOI:** 10.3390/jof9040398

**Published:** 2023-03-24

**Authors:** Ali Nuh, Newara Ramadan, Lisa Nwankwo, Jackie Donovan, Brijesh Patel, Anand Shah, Sujal R. Desai, Darius Armstrong-James

**Affiliations:** 1Laboratory Medicine, Department of Microbiology, Royal Brompton Hospital, Guy’s and St Thomas’ NHS Foundation Trust, London SW3 6NP, UK; 2Pharmacy Department, Royal Brompton Hospital, Guy’s and St. Thomas’ NHS Foundation Trust, London SW3 6NP, UK; 3Department of Chemical Pathology, Royal Brompton Hospital, Guy’s and St. Thomas’ NHS Foundation Trust, London SW3 6NP, UK; 4Division of Anaesthetics, Pain Medicine, and Intensive Care, Department of Surgery and Cancer, Centre for Haematology, Department of Immunology and Inflammation, and 5 National Heart and Lung Institute, Imperial College London, London SW3 6LY, UK; 5MRC Centre of Global Infectious Disease Analysis, Department of Infectious Disease Epidemiology, School of Public Health, Imperial College London, London SW7 2AZ, UK; 6Imaging Department, Royal Brompton Hospital, Guy’s and St. Thomas’ NHS Foundation Trust, London SW3 6NP, UK; 7National Heart and Lung Institute, Faculty of Medicine, Imperial College London, London SW3 6LY, UK; 8Margaret Turner-Warwick Centre for Fibrosing Lung Disease, Faculty of Medicine, Imperial College London, London SW7 2AZ, UK; 9Department of Respiratory Medicine, Royal Brompton Hospital, Guy’s and St Thomas’ NHS Foundation Trust, London SE1 7EH, UK; 10MRC Centre for Molecular Bacteriology and Infection, Department of Infectious Diseases, Imperial College London, London SW7 2AZ, UK

**Keywords:** *Aspergillus species*, pulmonary aspergillosis, ECMO, incidence, outcome, COVID-19

## Abstract

Background: The incidence and outcome of pulmonary aspergillosis in coronavirus disease (COVID-19) patients on extracorporeal membrane oxygenation (ECMO) are unknown and have not been fully addressed. We investigated the incidence, risk factors and outcome of pulmonary aspergillosis in COVID-19 ECMO patients. In addition, the diagnostic utility of bronchoalveolar lavage fluid and CT scans in this setting were assessed. Methods: We conducted a retrospective study on incidence and outcome of pulmonary aspergillosis in COVID-19 ECMO patients by reviewing clinical, radiological, and mycological evidence. These patients were admitted to a tertiary cardiothoracic centre during the early COVID-19 surge between March 2020 and January 2021. Results and measurements: The study included 88 predominantly male COVID-19 ECMO patients with a median age and a BMI of 48 years and 32 kg/m^2^, respectively. Pulmonary aspergillosis incidence was 10% and was associated with very high mortality. Patients with an Aspergillus infection were almost eight times more likely to die compared with those without infection in multivariate analysis (OR 7.81, 95% CI: 1.20–50.68). BALF GM correlated well with culture results, with a Kappa value of 0.8 (95% CI: 0.6, 1.0). However, serum galactomannan (GM) and serum (1–3)-β-D-glucan (BDG) lacked sensitivity. Thoracic computed tomography (CT) diagnostic utility was also inconclusive, showing nonspecific ground glass opacities in almost all patients. Conclusions: In COVID-19 ECMO patients, pulmonary aspergillosis incidence was 10% and associated with very high mortality. Our results support the role of BALF in the diagnosis of pulmonary aspergillosis in COVID-19 ECMO patients. However, the diagnostic utility of BDG, serum GM, and CT scans is unclear.

## 1. Introduction

Critically ill patients with viral pneumonia are at greater risk for pulmonary aspergillosis. In recent years, a growing body of literature has examined invasive aspergillosis super-infection in non-classically immunocompromised patients with severe influenza. In these studies, the influenza virus was identified as an independent risk factor for invasive aspergillosis and the clinicopathological characteristics of influenza-associated pulmonary aspergillosis (IAPA) were defined [1,2].

During the coronavirus disease 2019 (COVID-19) pandemic, a syndrome like IAPA was reported in critically ill patients with COVID-19. Several studies examined the relationship between COVID-19 and *Aspergillus* coinfection and reported a highly variable incidence of invasive aspergillosis in COVID-19 ARDS patients who were previously not known to be immunocompromised [3,4,5]. 

Invasive aspergillosis in these critically ill COVID-19 patients resulted in a change in the natural history of this disease and was associated with high mortality [4].

Most studies on COVID-19-associated pulmonary aspergillosis concentrate on mechanically ventilated patients by using upper respiratory tract samples such as tracheal aspirate, which have not been validated for galactomannan testing. This is due to the scarcity of bronchoalveolar fluid (BALF), which was often not collected because of the risk of nosocomial transmission of SARS-CoV-2 infection to health-care workers [6]. BALF is a benchmark specimen for diagnosing aspergillosis [7,8]. Furthermore, the extent of the incidence and outcome of pulmonary aspergillosis in COVID-19 patients on extracorporeal membrane oxygenation (ECMO) is unknown and has not been addressed. ECMO is used when conventional therapies such as intubation are exhausted, and these patients are often the most critically ill and therefore likely to be at risk of fungal infection, including *Aspergillus* spp.

In this retrospective study, we examined the diagnosis, incidence, risk factors and outcome of pulmonary aspergillosis in ECMO patients with full mycological work-up and high-resolution computerised chest tomography scans.

## 2. Material and Methods

### 2.1. Study Population and Case Definition

We retrospectively reviewed medical, radiological and microbiological records of COVID-19 ECMO patients admitted to Royal Brompton and Harefield Hospitals between March 2020 and January 2021. RBHT was one of the six dedicated ECMO centres in England during the COVID-19 pandemic. Data collected included demographics, comorbidities, radiological findings, and survival outcome. Only confirmed COVID-19 positive patients with full mycological analysis in bronchoalveolar (BALF: BALF culture; BALF and serum galactomannan and serum (1–3)-β-D-glucan (BDG)) determinations were included. Patients without mycological work-up were excluded. Data were extracted using the SAS^R^ software (SAS Institute, Cary, NC, USA) and exported into Excel. 

Probable COVID-19-associated pulmonary aspergillosis (CAPA) cases were defined according to recent consensus criteria by the European Confederation of Medical Mycology (ECMM) and The International Society for Human and Animal Mycology (ISHAM): COVID-19 patients needing ECMO care and with a BALF galactomannan index ≥ 1.0 and ground glass radiological appearance [9]. 

### 2.2. Microbiological Analysis

The presence of severe acute respiratory syndrome coronavirus 2 (SARS-CoV-2) was detected by real-time polymerase chain reactions (genesig^R^ Real-Time PCR COVID-19, Primerdesign, York, UK) using combined nasal and pharyngeal swabs. 

All patients were screened for pulmonary aspergillosis by routinely performing culture and galactomannan analysis on bronchoalveolar lavage fluid (BALF) at least twice weekly. In addition, clotted blood was tested for GM and Beta D-glucan. Each BALF sample was inoculated onto Sabouraud dextrose chloramphenicol agar (Biomerieux, culture media) and incubated at 37 °C for two days and at 30 °C for another five days. *Aspergillus* spp. were identified by culture characteristics. 

Galactomannan analysis on BALF and the serum was performed by sandwich enzyme-linked immunosorbent assay (Platelia^TM^ Bio-Rad, Watford, UK) as per the manufacturer’s instruction. The mycological criteria for aspergillosis were based on either the growth of *Aspergillus* species or a BALF galactomannan index (GMI) ≥1.0 or serum GMI ≥ 0.5. (1–3)-β-D-glucan (BDG) was detected using the Fungitell^®^ assay (Associates of Cape Cod, Liverpool, UK), using a cutoff of 80 pg/mL. 

### 2.3. Radiology

All patients underwent thoracic high-resolution computed tomography (HRCT) according to ECMO standard of care protocols. Radiology images were interpreted and scored by a blinded senior radiologist for radiological features associated with pulmonary aspergillosis.

### 2.4. Data Analysis

Data were analysed using IBM SPSS statistics for Windows, version 27.0 (IBM Corp., Armonk, NY, USA). Categorical variables were summarised with frequencies and percentages. Mann-Whitney U or χ^2^ tests were used to compare differences between survivors and non-survivors, in the case of quantitative and nominal variables (Table 1). To assess the risk factors that may be independently associated with aspergillosis, binary logistics regression was performed. Relevant variables such as baseline demographics, treatment and clinical characteristics were entered into multivariate regression models. Cohen’s Kapp was used to investigate the BALF diagnostic utility and its concordance with fungal culture. 

## 3. Results

After data sorting and deduplication, 88 patients met the inclusion criteria: positive SARS-2 PCR, receiving extracorporeal membrane oxygenation (ECMO) treatment and having bronchoalveolar lavage fluid (BALF) culture, BALF and serum galactomannan and serum (1–3)-β-D-glucan (BDG) determinations. Eight patients were excluded for the lack of mycological work-up. The mean age and body mass index were 48.1 (±9.26) and 31.9 (±7.71), respectively. A total of 73% (64) of the patients were male, and the most common comorbidity was hypertension, followed by diabetes and respiratory disease—predominantly asthma (Table 1). 

In this cohort, probable COVID-19-associated pulmonary aspergillosis (CAPA) incidence was 10%, with a 67% mortality rate. Nine patients were diagnosed with CAPA, and Table 2 shows the clinical and mycological characteristics of these patients. In bivariate analysis, CAPA was significantly associated with advanced age (*p* = 0.009) and the length of stay on ECMO (*p* < 0.001) (Table 1). Furthermore, overall mortality was 21% (18/88). Survival on ECMO treatment was only significantly associated with CAPA infection (*p* < 0.001) and days on ECMO stay (*p* = 0.003) (Table 3). 

In this cohort, we also investigated independent risk factors for CAPA and survival on ECMO treatment by performing multivariate analysis. All variables in Table 1 and Table 3 were used in the binary regression model. Days on EMCO was the only independent risk factor for CAPA infection. One extra day on ECMO was associated with an almost 8% increase in CAPA infection acquisition (OR 1.08, CI 95%: 1.03–1.13) (Table 4). As for survival on ECMO, CAPA was the only independent risk factor. Patients with CAPA were almost eight times more likely to die during ECMO treatment compared with those patients without CAPA infection (OR 7.81, 95% CI: 1.20–50.68) (Table 5). 

Furthermore, performance of BALF galactomannan (GM), serum GM, serum BDG and radiology in the diagnosis of probable CAPA was assessed. Patients diagnosed with probable CAPA exhibited a heavy burden of *Aspergillus* culture isolates in BALF. The BALF samples were repeatedly positive for *Aspergillus species* culture and biomarkers (Table 2). Culture, galactomannan and BDG were detected simultaneously in three samples. Six samples were also BALF GM and culture positive, growing predominantly *Aspergillus fumigatus*. BALF GM correlated well with culture with a Kappa value of 0.8 (95% CI: 0.6, 1.0).

Serum GM, serum BDG and radiology lacked diagnostic sensitivity. All patients were serum GM negative, and only three patients were beta D-glucan positive, in whom culture and BALF GM were simultaneously detected as well. CAPA-positive patient characteristics are summarised in Table 2. Almost all patients presented with nonspecific radiology showing ground glass opacities; only one patient had nodular and cavitating lesions. The radiology scores of patients with CAPA and without CAPA are presented in Table 6.

## 4. Discussion

To the best of our knowledge, the incidence and outcome of COVID-19-associated pulmonary aspergillosis (CAPA) in patients on extracorporeal membrane oxygenation (ECMO) are unknown. For the first time, in this retrospective study, we report a CAPA incidence of 10% and a mortality of 67% in ECMO patients who were continuously screened for fungal superinfection using bronchoalveolar lavage fluid.

Our results contrast with those of Fekkar et.al., who found no CAPA infection among ECMO patient in their cohort [10]. However, our incidence is close to figures reported for putative invasive pulmonary aspergillosis in non-COVID-19 patients receiving ECMO care. Rodriguez-Goncer et al. reported an incidence of 7% in this setting [11]. These researchers also reported a mortality rate of 80%, which is consistent with our observations.

ECMO patients have severe critical illnesses and are at risk of fungal infection, including *Aspergillus* spp. In addition, diagnosis of invasive pulmonary aspergillosis (IPA) is challenging. In this cohort, we continuously monitored for fungal superinfection by performing bronchoalveolar fluid (BALF) cultures and galactomannan as well as serum galactomannan and beta D-glucan. As previously reported by Koehler et al., we found BALF galactomannan (GM) diagnostically useful [3]. BALF GM correlated well with culture; six out of nine of GM positive cases were also culture positive, with a Kappa value of 0.8 (95% CI: 0.6, 1.0). This contrasted with serum GM, beta D-glucan and high-resolution computerised scanning, which lacked diagnostic sensitivity. All samples were serum GM negative, and only three patients were beta D-glucan positive.

As for the radiology results, most of the patients presented with nonspecific ground glass opacities; only one patient had nodular and cavitating lesions. Therefore, this study indicates that CAPA is not associated with specific radiological features in high-resolution computerised scanning. Similar observations have been reported by other researchers [11,12].

In total, nine patients were diagnosed with probable/putative CAPA. Of these, only three survived during ECMO treatment. The overall survival rate of patients with aspergillosis significantly differed from that of those without the disease (*p* < 0.001). In multivariate analysis, patients with CAPA were almost eight times less likely to survive compared to those without the disease. We specifically assessed whether co-morbidities known to increase risk of death from COVID-19 were contributory; however, this was not the case.

Furthermore, the length of EMCO stay was associated with development of CAPA infection. In multivariate analysis, only the length of ECMO was an independent risk factor of CAPA, where one extra day stay in ECMO was associated with an 8% increase in CAPA infection (OR 1.08, 95% CI: 1.03, 1.13). This concurs with previous results from Van Biesen et al. in critically ill COVID-19 patients with CAPA [13]. This may have clinical implications. Although the role of antifungal prophylaxis is understudied and unclear, it may be useful to consider length of stay on ECMO and the age prior to initiation of this therapy.

In conclusion, in a systematically characterised cohort of severe COVID-19 ECMO patients undergoing protocolised HRCT scanning and mycological surveillance, we observed a pulmonary aspergillosis incidence of 10%, which was associated with a very high mortality that was independent of other mortality risk factors. Our results support the role of BALF in the diagnosis of pulmonary aspergillosis in ECMO patients. However, they also show that HRCT scanning is not useful for the diagnosis of CAPA, as specific features are not observed. In addition, ages and the length of ECMO stay were found to be associated with higher risk of CAPA infection.

## 5. Limitations

The primary limitations of the study are that it is retrospective and single-centred. A further limitation was the lack of biopsy-based confirmation of pulmonary aspergillosis, as this procedure is very high risk in ECMO. Another limitation of this study is that the COVID-19 knowledge and guidance have evolved, and our findings might not be generalisable.

## Figures and Tables

**Table 1 jof-09-00398-t001:** Baseline characteristics of CAPA and non-CAPA patients.

Characteristics	Total Sample	CAPA (*n* = 9)	Non-CAPA (*n* = 79)	*p*-Value
Sex, Male, *n* (%)	64 (73)	7 (78)	57 (72)	0.72
Age (Years), Mean (SD)	48.1 (9.26)	55.6 (2.6)	47.2 (9.4)	0.009
ECMO Days, Mean (SD)	25.9 (19.4))	56 (26.9)	22.5 (15.2))	<0.001
BMI (Kg/m^2^)	31.9 (7.71)	32.4 (7.4)	31.8 (7.8)	0.84
Hypercholesterolaemia, *n* (%)	13 (15)	1 (11)	12 (15)	0.73
Hypertension, *n* (%)	33 (38)	3 (33)	30 (38)	0.78
Diabetes Mellitus, *n* (%)	21 (24)	2 (22)	19 (24)	0.90
Asthma & COPD, *n* (%)	22 (25)	2 (22)	20 (25)	0.84
Corticosteroid before COVID-19, *n* (%)	11 (13)	1 (11)	10 (13)	0.89
Tocilizumab Treatment, *n* (%)	8 (9)	8 (89)	0	0.32

Data are presented as *n* (%). Values are presented as mean and standard deviation. Abbreviations: BMI (Body Mass Index), ECMO (Extracorporeal Membrane Oxygenation), CAPA (COVID-19 Associated Pulmonary Aspergillosis.

**Table 2 jof-09-00398-t002:** Clinical and mycological characteristics of the nine patients with COVID-19-associated pulmonary aspergillosis.

Characteristic	Patient 1	Patient 2	Patient 3	Patient 4	Patient 5	Patient 6	Patient 7	Patient 8	Patient 9
Sex	M	M	M	M	M	M	M	F	F
Age	56	53	58	57	58	58	56	51	55
BMI	21.91	45.2	27.26	28.01	26.99	38.74	32.14	39.52	31.14
Comorbidities	None	None	HTN; Diabetes	HTN; Diabetes	HTN/Asthma	None	None	None	None
Steroid therapy before COVID-19	No	No	No	No	No	No	No	Yes	No
Antifungal treatment	Voriconazole	Voriconazole	Voriconazole	Voriconazole	Voriconazole	Voriconazole	Voriconazole	Voriconazole	Voriconazole
BAL GM index	>3.5	>3.5	>3.5	2.51	5.070	2.278	2.144	1.0	>3.5
Aspergillus Culture	*A. fumigatus*	*A. fumigatus*	*A. fumigatus*	*A. fumigatus*	No growth	*A. flavus*	No growth	No growth	*A. fumigatus*
Serum GM results	Negative	Negative	Negative	Negative	Negative	Negative	Negative	Negative	Negative
Beta D-Glucan	Negative	>500 pg/mL	245 pg/mL	Negative	Negative	118	Negative	Negative	Negative
Radiological findings	Non-specific	Non-specific	Non-specific	Non-specific	Nodules and cavitation	Non-specific	Non-specific	Non-specific	Non-specific
Outcome	Survivor	Survivor	Non-Survivor	Non-Survivor	Non-Survivor	Non-Survivor	Non-Survivor	Survivor	Non-Survivor

HTN (hypertension), GM (galactomannan), M (male), F (female), BMI (Body Mass Index).

**Table 3 jof-09-00398-t003:** Baseline characteristics of non-survivor (death) and survivor patients.

Characteristics	Total Sample	Non-Survivor (*n* = 18)	Survivor (*n* = 70)	*p*-Value (95% CI)
Sex, Male, *n* (%)	64 (73)	15 (83)	49 (70)	0.26
Age (Years), Mean (SD)	48.1 (9.26)	53.8 (7.19)	46.6 (9.21)	0.003
ECMO Days, Mean (SD)	25.9 (21.6	35.1 (28.09)	23.6 (15.95)	0.11
BMI (Kg/m^2^)	31.9 (7.71)	30.7 (6.23)	32.2 (8.06)	0.46
CAPA, *n* (%)	9 (10)	6 (33)	3 (4)	<0.001
Hypercholesterolaemia, *n* (%)	13 (15)	3 (17)	10 (14)	0.82
Hypertension, *n* (%)	33 (37)	6 (33)	27 (39)	0.68
Diabetes Mellitus, *n* (%)	21 (24)	3 (17)	18 (26)	0.42
Asthma & COPD, *n* (%)	22 (25)	5 (28)	17 (24)	0.76
Corticosteroid before COVID-19, *n* (%)	11 (13)	1 (6)	10 (14)	0.32
Tocilizumab Treatment, *n* (%)	8 (9)	3 (17)	5 (7)	0.21
Antifungal Treatment, *n* (%)	61 (69)	15 (83)	46 (66)	0.15

**Table 4 jof-09-00398-t004:** Risk factors for CAPA during ECMO treatment in multivariate analysis.

Multivariate Analysis	Odds Ratio (95% CI)	*p*-Value
Sex	1.209 (0.15–9.76)	0.86
Age (Years)	1.11 (0.98–1.25)	0.52
ECMO Days	1.08 (1.03–1.13)	**0.003**
BMI (Kg/m^2^)	2.79 (0.17–45.12)	0.47
Hypercholesterolaemia	1.08 (0.081–14.28)	0.96
Hypertension	0.26 (0.034–1.97)	0.19
Diabetes Mellitus	0.63 (0.088–4.47)	0.64
Asthma & COPD	0.55 (0.062–4.47)	0.62
Corticosteroid before COVID-19	0.55 (0.062–4.47)	0.62
Tocilizumab Treatment	na	

**Table 5 jof-09-00398-t005:** Risk factors for non-survivors (death) in COVID-19 patients on ECMO treatment.

Multivariate Analysis	Odds Ratio (95% CI)	*p*-Value
Sex	1.68 (0.33–8.57)	0.54
Age (Years)	4.11 (0.44–38.26)	0.214
ECMO Days	1.28 (0.29–5.73)	0.75
BMI (Kg/m^2^)	0.82 (0.14–4.68)	0.82
CAPA	7.81 (1.20–50.68)	0.031
Hypercholesterolaemia	1.84 (0.31–10.73)	0.50
Hypertension	0.69 (0.17–2.77)	0.60
Diabetes Mellitus	0.33 (0.058–2.82)	0.20
Asthma & COPD	2.13 (0.42–10.64)	0.36
Corticosteroid before COVID-19	0.25 (0.016–3.96)	0.33
Tocilizumab Treatment	0.39 (0.083–1.79)	0.223
Antifungal treatment	1.93 (0.37–10.20)	0.44

**Table 6 jof-09-00398-t006:** Observed radiological features in CAPA and non-CAPA patients.

	CAPA Patients (*n* = 9)	Non-CAPA Patients (*n* = 79)
Nodules, *n* (%)	1 (11)	15 (19)
Cavitation, *n* (%)	1 (11)	4 (5)
Pleural thickening, *n* (%)	0	0
Bronchial thickening, *n* (%)	0	3 (4)
Airway plugging, *n* (%)	0	0

## Data Availability

This study didn’t report any data; anonymized data is stored in a secure NHS trust server.

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
