# Peer review of "COVID-19 Associated Pulmonary Aspergillosis in Patients on Extracorporeal Membrane Oxygenation Treatment—A Retrospective Study"

_jof, 2023, doi:10.3390/jof9040398_

Round 1

Reviewer 1 Report

Nuh and colleagues performed a retrospective observational study about CAPA in COVID-19 patients on ECMO treatment.

Data about CAPA in this specific subset of patients are lacking, so this study is potentially useful to better describe this invasive fungal infection.

However, there are some issues about this paper and my main concerns are the study design and the way data are reported.

Authors excluded from the analysis patients that didn’t have full mycological workup and patients who received antimicrobial that may cause false positive GM results, such as pip/taz. Authors should report details about the screening process, i.e. how many patients were excluded for each of these criteria?  Many patients with critical COVID-19 were likely to receive broad spectrum antibiotics such as pip/taz, especially during the first two pandemic waves. Furthermore, using the ECMM/ISHAM algorithm, a single positive mycological test is sufficient to diagnose CAPA in critical COVID-19 patients. Were the patients you excluded due to incompleteness of the mycological work-up not tested at all or were they partially tested, and the results were negative? Please explain.

The focus of the paper should be CAPA on ECMO patients (incidence and risk factors), thus I would make a comparison only between CAPA and non-CAPA patients (I would reframe Table 1). After quickly describing the entire cohort of patients, the Results section should focus on CAPA (incidence, patients characteristics etc…, i.e. from line 143). Line 132-139 should be reported later.

Line 162: you cite Table 3 for the univariate analysis, but Table 3 shows the clinical characteristics of CAPA patients. Probably, the univariate analysis table (CAPA vs. non-CAPA) is missing. You should report more details about univariate analysis.

Line 164-165: you cite the results of multivariate analysis, but we do not known which variables were used for the model. Please explain.

Line 74-77: please move this part in section “Material and methods”.

Line 89: I think you mean “probable CAPA”, not putative. ECMM/ISHAM consensus definition is about proven, probable or possible CAPA.

Line 148: How was this coefficient calculated? Is this the Cohen’s Kappa? If you perform this measurement, this should be explained in the Data analysis/Statistical Methods section.

Author Response

REVIEWER 4:

Nuh and colleagues performed a retrospective observational study about CAPA in COVID-19 patients on ECMO treatment.

  1. Data about CAPA in this specific subset of patients are lacking, so this study is potentially useful to better describe this invasive fungal infection.

Response: thanks,  results section rewritten, and we included extra table showing baseline characteristics of CAPA patients and Non-CAPA patients. 

  1. However, there are some issues about this paper and my main concerns are the study design and the way data are reported.

Authors excluded from the analysis patients that didn’t have full mycological workup and patients who received antimicrobial that may cause false positive GM results, such as pip/taz. Authors should report details about the screening process, i.e. how many patients were excluded for each of these criteria?  Many patients with critical COVID-19 were likely to receive broad spectrum antibiotics such as pip/taz, especially during the first two pandemic waves. Furthermore, using the ECMM/ISHAM algorithm, a single positive mycological test is sufficient to diagnose CAPA in critical COVID-19 patients. Were the patients you excluded due to incompleteness of the mycological work-up not tested at all or were they partially tested, and the results were negative? Please explain.

Response: patients with CAPA infection were repeatedly positive and no need existed to suspect false positive due to the use of Piperacillin-tazobactam.  However, 8 patients without mycological work up were excluded as BAL was not collected. 

  1. The focus of the paper should be CAPA on ECMO patients (incidence and risk factors), thus I would make a comparison only between CAPA and non-CAPA patients (I would reframe Table 1). After quickly describing the entire cohort of patients, the Results section should focus on CAPA (incidence, patients characteristics etc…, i.e. from line 143). Line 132-139 should be reported later.

Response: Thanks for the suggestion; noted and the results section rewritten and re-organised with an extra table showing baseline characteristics of CAPA patients and non-CAPA patients.  . 

  1. Line 162: you cite Table 3 for the univariate analysis, but Table 3 shows the clinical characteristics of CAPA patients. Probably, the univariate analysis table (CAPA vs. non-CAPA) is missing. You should report more details about univariate analysis.
  2. Line 164-165: you cite the results of multivariate analysis, but we do not know which variables were used for the model. Please explain.

Response: all explanatory variables in table 1 were entered into the regression model.  Clarified in the methods and results sections. 

  1. Line 74-77: please move this part in section “Material and methods”.

Response: noted and moved to method section. 

  1. Line 89: I think you mean “probable CAPA”, not putative. ECMM/ISHAM consensus definition is about proven, probable or possible CAPA.

Response: changed and correct term used. 

  1. Line 148: How was this coefficient calculated? Is this the Cohen’s Kappa? If you perform this measurement, this should be explained in the Data analysis/Statistical Methods section.

Response: Cohen’s Kappa was used to investigate BALF diagnostic utility and concordance with fungal culture.  This is now mentioned in the data analysis section. 

Reviewer 2 Report

Dear Editor,

Thank you very much for the opportunity to review this manuscript. In it, authors describe ECMO as a risk factor for CAPA development in patients at risk. The topic is very interesting, as these patients present a per se baseline increased risk for a fatal outcome as compared to non-ECMO ones. There are some aspects that after consideration by authors, could improve the quality of the manuscript.

TITLE

-          Since data are slightly “old” from a pandemic chronological perspective, the manuscript could benefit from stating in the title that these are “historical” data.

ABSTRACT

-          Line 27-28: “has not been addressed”. Despite long series on ECMO are not available, this is not complete accurate. There are indeed some reports (i.e., Helleberg et al. CMI 2020, and there might be more)

-          “pulmonary aspergillosis incidence was 10%” If possible, provide the ward (if any)

-          “Patients with Aspergillus infection were 11.2 times more likely” Please, provide the adjusted ratio, the crude one might be biased. If it is the adjusted, please specify

INTRODUCTION

-          Authors may want to include “Blot S, Taccone F, Van Abeele A, Bulpa P, Meersseman W, Brusselaers N, Dimopoulos G, Paiva JA, Misset B, Rello J, Vandewoude K, Vogelaers D, AspICU Study Investigators A clinical algorithm to diagnose invasive pulmonary aspergillosis in ICU patients. Am J Respir Crit Care Med. 2012;186:56–64. doi: 10.1164/rccm.201111-1978OC.” As reference for IAPA considering the work of Blot et al generating a definition algorithm widely used in the literature, even at the beginning of CAPA

-          “reported a highly variable incidence of invasive aspergillosis” Manuscript quality may improve from providing numbers to that incidence. Salmanton-García et al. Emerging Infectious Diseases 2021 might be a good option.

-          “which was often not collected because of risk of nosocomial transmission of SARS-CoV-2 infection to health-care workers” might be accompanied by Koehler P, Cornely OA, Kochanek M. Bronchoscopy safety precautions for diagnosing COVID-19 associated pulmonary aspergillosis-A simulation study. Mycoses. 2021 Jan;64(1):55-59. doi: 10.1111/myc.13183. Epub 2020 Sep 26. PMID: 32918497.

-          Authors may want to transfer “RBHT is one of the six dedicated ECMO centre in England during COVID-19 pandemic.” to Methods

-          Third paragraph is a single sentence. Inclusion to a larger paragraph could be welcome

METHODS

-          Authors may want to keep the coherence when mentioning the companies supplying the different diagnostic kits, software, etc

RESULTS

-          Did authors think of providing an explanatory figure for “After data sorting and de-duplication, 88 patients met the inclusion criteria”?

-          Authors may prefer to use median and interquartile range in case their continuous variables are non-normally distributed

-          Authors may want to include the n and the % (together) after mentioning each factor in the descriptive statistics

-          Line 128: Authors may want to review and recheck all the abbreviations used “SARS-2 PCR”

-          Did authors consider using Cox regression for mortality instead of logistic regression, as the first is considering the time of observation?

-          Authors may want to avoid demonstrative adjectives (these, this, that…) for the sake of clarity and use the respective term behind, even if repetitive

-          Table 3 might benefit if authors would provide length of antifungal treatment, whether there was any antifungal prophylaxis…

-          Table 5 could improve providing the percentages

DISCUSSION

-          Authors may consider to update their references as some of them are “old” for the topic in discussion. Nevertheless, it is also true that the dates of data collection for this manuscript places the authors in not the best situation: historical results to be compared to historical data, presume?

-          Authors may want to compare their results with other experiences or provide an interpretation (lines 195 to 200)

-          Would authors say that the fact that their results are historical and not recent are also a limitation?

-          In authors’ opinion, what is more relevant: conclusions or limitations? If the first, they may want to make the conclusions the conclusive paragraph and the final message of their manuscript

Author Response

Response to reviewer 1.

We thank the reviewer for reviewing our manuscript and for the detailed comments and suggestions.  We have revised the manuscript in line with these comments and suggestions. 

REVIEWER:

Dear Editor,

Thank you very much for the opportunity to review this manuscript. In it, authors describe ECMO as a risk factor for CAPA development in patients at risk. The topic is very interesting, as these patients present a per se baseline increased risk for a fatal outcome as compared to non-ECMO ones. There are some aspects that after consideration by authors, could improve the quality of the manuscript.

TITLE

  1. Since data are slightly “old” from a pandemic chronological perspective, the manuscript could benefit from stating in the title that these are “historical” data.

Title revised and the word retrospective added to highlight the historical nature of the data

ABSTRACT

  1. Line 27-28: “has not been addressed”. Despite long series on ECMO are not available, this is not complete accurate. There are indeed some reports (i.e., Helleberg et al. CMI 2020, and there might be more).

Response: thanks, wording changed to reflect this.

  1. “pulmonary aspergillosis incidence was 10%” If possible, provide the ward (if any)

Response: dedicated part of adult intensive care unit

  1. “Patients with Aspergillus infection were 11.2 times more likely” Please, provide the adjusted ratio, the crude one might be biased. If it is the adjusted, please specify.

Response: corrected and adjusted OR provided- multivariate analysis was performed. 

INTRODUCTION

  1. Authors may want to include “Blot S, Taccone F, Van Abeele A, Bulpa P, Meersseman W, Brusselaers N, Dimopoulos G, Paiva JA, Misset B, Rello J, Vandewoude K, Vogelaers D, AspICU Study Investigators A clinical algorithm to diagnose invasive pulmonary aspergillosis in ICU patients. Am J Respir Crit Care Med. 2012;186:56–64. doi: 10.1164/rccm.201111-1978OC.” As reference for IAPA considering the work of Blot et al generating a definition algorithm widely used in the literature, even at the beginning of CAPA.

Response: thanks, this useful paper was acknowledged and referenced

  1. “Reported a highly variable incidence of invasive aspergillosis” Manuscript quality may improve from providing numbers to that incidence. Salmanton-García et al. Emerging Infectious Diseases 2021 might be a good option.

Response: thanks, this useful paper was acknowledged and referenced

  1. “which was often not collected because of risk of nosocomial transmission of SARS-CoV-2 infection to health-care workers” might be accompanied by Koehler P, Cornely OA, Kochanek M. Bronchoscopy safety precautions for diagnosing COVID-19 associated pulmonary aspergillosis-A simulation study. Mycoses. 2021 Jan;64(1):55-59. doi: 10.1111/myc.13183. Epub 2020 Sep 26. PMID: 32918497.

Response: thanks this useful paper was acknowledged and referenced

  1. Authors may want to transfer “RBHT is one of the six dedicated ECMO centre in England during COVID-19 pandemic.” to Methods

Response: sentence moved to methods section. 

  1. Third paragraph is a single sentence. Inclusion to a larger paragraph could be welcome.

Response: restructured

METHODS

  1. Authors may want to keep the coherence when mentioning the companies supplying the different diagnostic kits, software, etc

Response: thanks, noted and acted upon.

RESULTS

  1. Did authors think of providing an explanatory figure for “After data sorting and de-duplication, 88 patients met the inclusion criteria”?

Response: explanatory variables were Gender, Age (Years), days patient spent on ECMO, BMI (Kg/m2), GM Index, CAPA diagnosis, Cholesterol, Hypertension, Diabetes Mellitus, respiratory co-morbidity (Asthma & COPD), Corticosteroid before COVID-19, Tocilizumab Treatment and Antifungal Treatment.

  1. Authors may prefer to use median and interquartile range in case their continuous variables are non-normally distributed.

Response: thanks, noted, the only variable that would need reporting median galactomannan (GM) variable, however, GM is substituted by CAPA in table 1.  Other variables such age and BMI were normally distributed and hence the mean with SD was reported.   

  1. Authors may want to include the n and the % (together) after mentioning each factor in the descriptive statistics.

Response: n and % included

  1. Line 128: Authors may want to review and recheck all the abbreviations used “SARS-2 PCR”

Response: corrected

  1. Did authors consider using Cox regression for mortality instead of logistic regression, as the first is considering the time of observation?

Response: tried but found multivariate regression more suitable for this data

  1. Authors may want to avoid demonstrative adjectives (these, this, that…) for the sake of clarity and use the respective term behind, even if repetitive.

Response: thanks, noted and replaced demonstrative adjectives where necessary. 

  1. Table 3 might benefit if authors would provide length of antifungal treatment, whether there was any antifungal prophylaxis…

Response: almost 70% of patients were on Echinocandin prophylaxis (mainly anidulafungin); this made no significant difference in the outcome (table 1).  When CAPA was diagnosed Voriconazole was started and continued during EMCO stay. 

  1. Table 5 could improve providing the percentages.

Response: included

DISCUSSION

  1. Authors may consider to update their references as some of them are “old” for the topic in discussion. Nevertheless, it is also true that the dates of data collection for this manuscript places the authors in not the best situation: historical results to be compared to historical data, presume?

Response: tried to update and literature search conducted; but to our knowledge there no structured COVID19-ECMO exclusive studies with full radiological and mycological investigations. 

  1. Authors may want to compare their results with other experiences or provide an interpretation (lines 195 to 200)

Response:

  1. Would authors say that the fact that their results are historical and not recent are also a limitation?

Response: yes we agree and include in the limitations.

  1. In authors’ opinion, what is more relevant: conclusions or limitations? If the first, they may want to make the conclusions the conclusive paragraph and the final message of their manuscript.

Response: conclusion high mortality of CAPA-ECMO patients and diagnostic utility of BALF galactomannan in this setting. 

Reviewer 3 Report

This is an interesting communication about 88 cases of patients with covid-19, aspergillosis and use of ECMO. Authors considered diagnosis, incidence, risk factors and outcome. I have some observations:

line 201, correct "ECMO". 

line 202, capitalize In 

Reference 7 is missing the year of publication

line 128, SARS-Cov2

I could not understand if mortality was 10% ou 21% among patients. Please explain

Author Response

REVIEWER 2:

This is an interesting communication about 88 cases of patients with covid-19, aspergillosis and use of ECMO. Authors considered diagnosis, incidence, risk factors and outcome. I have some observations:

  1. line 201, correct "ECMO". 

Response: corrected

  1. line 202, capitalize In 

Response: thanks, capitalised

  1. Reference 7 is missing the year of publication

Response: added the year of publication

  1. line 128, SARS-Cov2

Response: corrected the typo

  1. I could not understand if mortality was 10% ou 21% among patients. Please explain.

Response: explained; mortality of the whole cohort- all cause mortality was 21%

Reviewer 4 Report

In this article, Nuh, et al. performed a retrospective study describing the incidence and outcomes of COVID-19 associated pulmonary aspergillosis in patients on ECMO between March 2020 to January 2021. 88 subjects with COVID on ECMO were included, only 9 of which had evidence of CAPA. This data from a group that regularly screened for fungal infection with both serum and BAL surveillance testing is interesting and worthy of publication in the larger context of COVID-associated fungal infections. The authors also describe some potential risk factors for survival in COVID19 on ECMO, though much of this data has been previously described in the COVID literature. Some minor comments/revisions:

-Some of the data interpretation on diagnostic testing for CAPA in the outcomes/discussion might be slightly overstated in the context of the small sample size (only 9 patients with CAPA). This should be stated in the limitations.

-There have been other case reports and smaller studies describing Aspergillus infection in COVID patients on ECMO. Though perhaps not as structured as this study, it is often challenging to state that this is the 'first' study on CAPA in ECMO patients. (If maintaining this statement, need to make it clear that you mean only in the context of those 'who were continuously screening for fungal superinfection as stated in the latter part of this sentence in the discussion.') In fact, in the next sentence, the authors describe that their results contrast another report examining for CAPA in ECMO patients - would be helpful for the authors to describe what may have been different between their study and the one they are referencing, given the great disparity in data and conclusions.

-This manuscript covers a time period that was at an earlier stage in the COVID19 pandemic, when antiviral treatments were first evolving and CAPA was only being first described. Steroid doses were still being clarified, and tocilizumab was being more frequently used during this time period. Do the authors think the outcomes and incidence would be different in the more recent time period, now that COVID therapies have evolved? Do they think the incidence of CAPA would go down as ECMO is less frequently necessary for COVID treatment?

Author Response

REVIEWER 3:

In this article, Nuh, et al. performed a retrospective study describing the incidence and outcomes of COVID-19 associated pulmonary aspergillosis in patients on ECMO between March 2020 to January 2021. 88 subjects with COVID on ECMO were included, only 9 of which had evidence of CAPA. This data from a group that regularly screened for fungal infection with both serum and BAL surveillance testing is interesting and worthy of publication in the larger context of COVID-associated fungal infections. The authors also describe some potential risk factors for survival in COVID19 on ECMO, though much of this data has been previously described in the COVID literature. Some minor comments/revisions:

  1. -Some of the data interpretation on diagnostic testing for CAPA in the outcomes/discussion might be slightly overstated in the context of the small sample size (only 9 patients with CAPA). This should be stated in the limitations.

Response: note small positive number added to the limitations of the study.

  1. -There have been other case reports and smaller studies describing Aspergillus infection in COVID patients on ECMO. Though perhaps not as structured as this study, it is often challenging to state that this is the 'first' study on CAPA in ECMO patients. (If maintaining this statement, need to make it clear that you mean only in the context of those 'who were continuously screening for fungal superinfection as stated in the latter part of this sentence in the discussion.') In fact, in the next sentence, the authors describe that their results contrast another report examining for CAPA in ECMO patients - would be helpful for the authors to describe what may have been different between their study and the one they are referencing, given the great disparity in data and conclusions.

Response: thanks, clarified; first structure study with complete fungal and CT scan screen in ECMO patients

  1. -This manuscript covers a time period that was at an earlier stage in the COVID19 pandemic, when antiviral treatments were first evolving and CAPA was only being first described. Steroid doses were still being clarified, and tocilizumab was being more frequently used during this time period. Do the authors think the outcomes and incidence would be different in the more recent time period, now that COVID therapies have evolved? Do they think the incidence of CAPA would go down as ECMO is less frequently necessary for COVID treatment?

Response: still incidence of CAPA-ECMO is not fully clarified.  However, improved COVID19 treatment may have reduce the number of patients that require ECMO treatment and CAPA-ECMO incidences. 

Round 2

Reviewer 1 Report

Just two minor revisions.

Line 75-77: move to Materials and methods please

Table 1: I think you mean hypercholesterolemia and not "cholesterol"

Author Response

Thanks for the useful comments: 

  1. line 57-77 move to methods section.  
  2. Hypercholesterolaemia term was used instead of cholesterol